# Effect of Nanosilica on the Strength and Durability of Cold-Bonded Fly Ash Aggregate Concrete

**Xiuzhong Peng** [1,2] 🆔, **Qinghua Wang** [2,3] **and Jing Wu** [2,4],*

1   School of Civil Engineering, Southeast University, Nanjing 211189, China; pengxiuzhong163@163.com
2   College of Construction Engineering, Jilin University, Changchun 130021, China
3   Shanghai Geological Engineering Exploration (Group) Co., Ltd., Shanghai 200072, China
4   Beijing Institute of Architectural Design, Beijing 100045, China
*   Correspondence: wujing3@biad.com.cn

**Abstract:** Cold-bonded Fly Ash Aggregate (CFAA), as an alternative to natural coarse aggregates, can prepare more lightweight, economical, and sustainable concrete. However, CFAA concrete has insufficient durability, which hinders its application in a salt-corrosion environment. Nanosilica (NS) has an advantage of high activity and is generally used as an efficient mineral admixture in engineering. This study aims to improve the strength and durability of CFAA concrete by incorporating NS. To this end, compression tests, splitting tensile tests, and microscopic analyses were performed to investigate the mechanical properties of the concrete containing different NS dosages. Subsequently, the dry–wet and freeze–thaw durability tests were conducted to evaluate the salt-corrosion resistance and the frost resistance in the water, $Na_2SO_4$ solution, and $Na_2CO_3$ solution. The results show the compressive and splitting tensile strength peak at 2 wt% NS dosage. In this instance, the concrete has an optimum microstructure and exhibits desirable salt-corrosion resistance in the late stage of dry–wet cycles. During freeze–thaw cycles, NS could improve the frost resistance of the concrete but scarcely diminished internal damage under sulfate attack. The study explores the long-term performance of NS-modified CFAA concrete, providing a simple and effective method to mitigate the concrete deterioration in a harsh environment.

**Keywords:** cold-bonded; fly ash aggregate concrete; nanosilica; mechanical properties; durability; microstructure





## 1. Introduction

Cold-bonded Fly Ash Aggregate (CFAA) enables the disposal of plentiful industrial wastes to diminish environmental pollution and achieves the production process with less energy consumption than the sintered method [1–3]. In engineering, this aggregate is gradually becoming an alternative to natural aggregates to produce lightweight concrete [4–6]. CFAA concrete possesses the potential to decrease dead loads, reduce component sizes, and save transportation costs, having wide application prospects in building materials and structural design [1,7].

CFAA concrete is characterized by an economical and ecofriendly material [8–10]. The production process of CFAA includes pelletization and curing, achieving the recycling of fly ash, which is a by-product of thermal power plants [11–13]. Pelletization means that water is sprayed on a mixture of fly ash and cement in a rotating disc to form almost spherical particles. Curing aims to allow these particles to reach the desired strength through hydration reaction and pozzolanic reaction at ambient temperature. The manufacture of CFAA is simple and efficient, and only requires a small amount of energy during the curing process [14–16]. The superiority in cost makes CFAA concrete more attractive than other lightweight aggregate concrete, such as vermiculite concrete [17], perlite concrete [18], and pumice concrete [19]. Since aggregate makes up at least 60% of concrete by volume, the use of CFAA in concrete would consume fly ash rapidly [20,21]. The decrease in the large

amount of the industrial waste contributes to avoiding the adverse effect on human health and nature [22].

Despite these advantages, CFAA concrete generally has a lower strength, owing to aggregate crushing and matrix cracking [23]. Ibrahim et al. [24] found CFAA concrete had weaker bond strength compared to ordinary concrete through reinforcement steel bar pull-out tests. Some studies focused on strengthening aggregates to improve concrete performance [25,26]. Patel et al. [27] took styrene butadiene rubber latex as a binder to manufacture fly ash aggregates, making the concrete obtain desirable compressive strength and transport properties. Yıldırım and Özturan [28] added polypropylene fibers to the aggregates so that the impact resistance of the concrete was enhanced. Moreover, some researchers pointed out that mineral admixtures can optimize the microstructure of a cementitious matrix and thus enhance the mechanical properties of CFAA concrete [29,30]. Joseph and Ramamurthy [31] advised that concrete could obtain a higher strength through replacing sand rather than cement with fly ash. Güneyisi et al. [32] reported that Silica Fume (SF) increased the compressive strength of self-compacting CFAA concrete, whereas fly ash might reduce the strength. Due to its high specific surface area, Nanosilica (NS) has better physical effects and chemical properties than conventional mineral admixtures [33,34]. Recent studies [35,36] discovered that NS with smaller particle sizes could accelerate cement hydration and have the higher pozzolanic activity to improve the mortar performance. Son et al. [37] revealed that the addition of NS decreased the generation rate of metastable hydrates, maintaining the long-term strength of calcium aluminate cement. With the development of nanotechnology, the use of NS in concrete has grown in recent years [38,39]. Wang et al. [40] demonstrated that NS-modified underwater concrete had higher compressive strength and lower permeability, contributing to resisting seawater attacks. Mukharjee and Barai [41] applied NS to recycled aggregate concrete to mitigate performance degradation due to the inherent defects of the aggregates. Farajzadehha et al. [42] found that the strength of plastic concrete increased with the growth of NS dosage through the uniaxial and triaxial compression tests. Murad et al. [43] studied the cyclic behavior of beam−column joints made with NS-modified concrete and discovered that the failure mode of the specimens was changed due to the incorporation of NS. Mashshay et al. [44] illustrated that NS could alleviate the postfire mechanical degradation of lightweight concrete.

Furthermore, the durability of CFAA concrete is a matter of concern, especially in a harsh environment. In saline soil areas, concrete suffers salt attacks and undergoes dry–wet and freeze–thaw cycles as precipitation and temperatures vary seasonally. As a result, the surface spallation and microcracks of the concrete can continually arise, leading to a dramatic reduction in performance. In terms of natural aggregate concrete, some advances in durability have been reported. Wang et al. [45] performed uniaxial compression tests to explore the concrete deterioration under the coupling of loads and wet–dry cycles. Liu et al. [46] stated that the physical and mechanical properties increased first and then decreased when the concrete was subjected to sulfate attacks and dry–wet cycles. Guo et al. [47] and Wang et al. [48] concluded that low dry–wet time ratio exacerbated the chemical erosion of sulfates and decreased the crystallization pressure. Zhou et al. [49] showed that salt attacks and initial load damage caused more strength loss during freeze–thaw cycles. Wang et al. [50] and Chen et al. [51] established freeze–thaw damage models to predict the damage degree of the concrete exposed to a saline environment. Yang et al. [52] believed that concrete deterioration would greatly accelerate under the coupling action of sulfate attacks, dry–wet cycles, and freeze–thaw cycles. However, the research on the durability of CFAA concrete is still in its infancy.

Present studies [45–52] are mostly related to the durability of natural aggregate concrete. It is difficult to directly apply these results to the durability improvement of lightweight concrete due to the differences in aggregates. Consequently, it is a considerable challenge to solve the deterioration of CFAA concrete in a salt-corrosion environment. To this end, it is necessary to find a simple and effective method to strengthen the matrix

performance, helping the concrete resist salt erosion. In this study, partial cement was substituted with NS to improve the long-term performance of CFAA concrete subjected to sulfate and carbonate attacks.

This study aimed to explore the mechanical properties and durability of NS-modified CFAA concrete. Compression tests and splitting tensile tests were conducted to find the optimum NS dosage. The failure morphologies and microstructure images of the specimens were observed to analyze the influence mechanism of NS. Subsequently, the dry–wet and freeze–thaw cycle tests were performed to evaluate the salt-corrosion resistance and the frost resistance in the water, $Na_2SO_4$ solution, and $Na_2CO_3$ solution.

The highlights of this work are summarized below:

1.  CFAA concrete had the optimum mechanical properties at 2% NS dosage.
2.  The porosity of CFAA concrete was reduced obviously by the moderate use of NS.
3.  NS improved the salt-corrosion resistance of CFAA concrete during dry–wet cycles.
4.  The desirable freeze–thaw resistance under salt attacks could be achieved by incorporating NS in CFAA concrete.

## 2. Materials and Methods

### 2.1. Raw Materials

#### 2.1.1. Cement

Ordinary Portland cement (Jilin Yatai Dinglu Cement Co., Ltd., Changchun, China) with a strength grade of 42.5 MPa was used in this experiment, conforming to the Chinese standard GB 175-2007 [53]. The properties and chemical composition of cement are listed in Table 1.

**Table 1.** Properties and chemical composition of cement, Silica Fume (SF), and Nanosilica (NS).

| Items | Cement | SF | NS |
|---|---|---|---|
| Properties | | | |
| Average particle size (um) | – | 0.1–0.3 | 0.02 |
| Specific gravity | 3.12 | 2.18 | – |
| Specific surface area ($m^2$/kg) | 358 | 18,000 | 200,000 |
| Loss on ignition (wt%) | 1.6 | 2.0 | 2.0 |
| Chemical composition (wt%) | | | |
| $SiO_2$ | 20.62 | 93.30 | >99.50 |
| $Al_2O_3$ | 5.14 | 0.73 | – |
| CaO | 65.07 | 0.85 | – |
| MgO | 0.87 | 1.21 | – |
| $Fe_2O_3$ | 3.96 | 0.49 | – |

#### 2.1.2. Mineral Admixture

The added mineral admixtures included NS (Jiangsu Tianxing New Material Co., Ltd., Huaian, China) and SF (Changchun Siao Technology Co., Ltd., Changchun, China). NS, a white powder with high dispersion, has PH values ranging from 7 to 9 and a loose bulk density of 100 kg/$m^3$. SF can be combined with NS to optimize the particle gradation of cementitious materials [54]. Table 1 also gives the properties and chemical composition of both SF and NS.

#### 2.1.3. Coarse Aggregate

CFAA (Gongyi Lanke Water Purification Material Co., Ltd., Zhengzhou, China) with particle sizes of 3–5 mm was chosen as the coarse aggregate, as shown in Figure 1a. In Figure 1b, the Scanning Electron Microscope (SEM) image of the aggregate indicated CFAA was full of Calcium-Silicate-Hydrate (C-S-H) gels. Figure 2a shows that the grading curve of the CFAA was within the boundaries of the nominal size grade (5–10 mm) according to the Chinese standard GB/T 17431.1-2010 [55]. It indicated that the particle gradation of

the CFAA met the grading requirements of a lightweight coarse aggregate. The physical properties of CFAA are tabulated in Table 2.

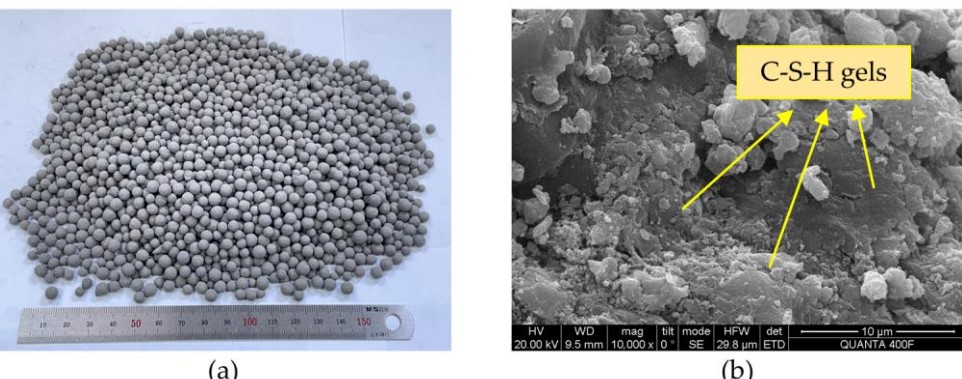

(a)                                                              (b)

**Figure 1.** Characteristics of Cold-bonded Fly Ash Aggregate (CFAA): (**a**) the CFAA with particle sizes of 3–5 mm; (**b**) the Scanning Electron Microscope (SEM) image of the aggregate.

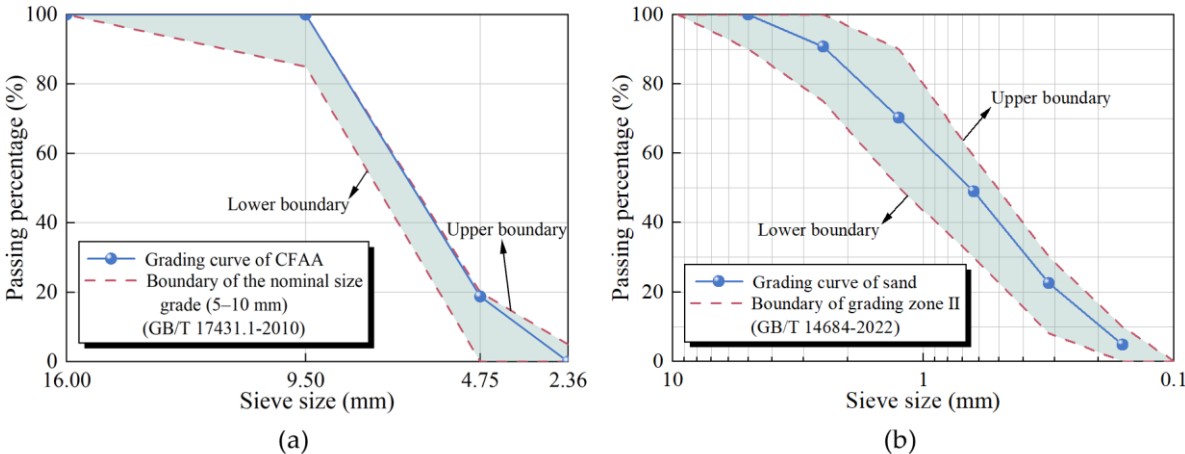

(a)                                                              (b)

**Figure 2.** Sieve test results of CFAA and sand: (**a**) grading curve of CFAA; (**b**) grading curve of sand.

**Table 2.** Physical properties of Cold-bonded Fly Ash Aggregate (CFAA) and sand.

| Items | CFAA | Sand |
|---|---|---|
| Bulk density (kg/m$^3$) | 920 | 1670 |
| Apparent density (kg/m$^3$) | 1582 | 2663 |
| Water absorption (wt%) | | |
| 1 h | 19 | — |
| 24 h | 20 | — |

### 2.1.4. Sand

The fine aggregate was river sand with a fineness modulus of 2.58, and its sieve test results are illustrated in Figure 2b. The results showed that the gradation curve of the sand was in the range of the gradation zone II specified by the Chinese standard GB/T 14684-2022 [56]. Table 2 also presents the physical properties of sand.

### 2.1.5. Water

Tap water (Changchun Water Co., Ltd., Changchun, China) was used to prepare fresh concrete and cure specimens.

### 2.1.6. Superplasticizer

Polycarboxylate superplasticizer, a transparent liquid with a water reduction rate of 25%, was applied to attain the desired workability of the fresh concrete.

### 2.2. *Mix Design*

The mix design of CFAA concrete was performed by the loose volume method based on the Chinese standard JGJ/T 12-2019 [57]. The target compressive strength of the concrete was set to 40 MPa. During mix design, the total cementitious materials, the water-to-binder ratio, and the loose volume rate of sand were taken as 435 kg/m$^3$, 0.35, and 35 vol%, respectively. For CFAA concrete without NS, the mix proportion was labeled as N0. The cementitious materials of N0 contained 93 wt% cement and 7 wt% SF. The previous studies [58–62] demonstrated that the mechanical properties of concrete can be improved significantly when the NS dosage is within the range from 1 wt% to 4 wt%. In this study, NS partially replaced cement and SF to prepare CFAA concrete with different NS dosages (1 wt%, 2 wt%, 3 wt%, and 4 wt%). The resulting mix proportions were labeled N1, N2, N3, and N4, respectively. The dosage of superplasticizer was determined as 1.5 wt% of the total cementitious materials. Table 3 lists the mix proportions of CFAA concrete.

**Table 3.** Mix proportions of CFAA concrete.

| Mix No. | NS Dosage (wt%) | Weight per Cubic Meter (kg/m$^3$) | | | | | | |
| --- | --- | --- | --- | --- | --- | --- | --- | --- |
| | | Cement | SF | NS | CFAA | Sand | Water | Superplasticizer |
| N0 | 0 | 404.55 | 30.45 | 0 | 717.6 | 701.4 | 152.25 | 6.525 |
| N1 | 1 | 400.50 | 30.15 | 4.35 | 717.6 | 701.4 | 152.25 | 6.525 |
| N2 | 2 | 396.46 | 29.84 | 8.70 | 717.6 | 701.4 | 152.25 | 6.525 |
| N3 | 3 | 392.41 | 29.54 | 13.05 | 717.6 | 701.4 | 152.25 | 6.525 |
| N4 | 4 | 388.37 | 29.23 | 17.40 | 717.6 | 701.4 | 152.25 | 6.525 |

### 2.3. *Specimen Preparation*

The preparation process of specimens is depicted in Figure 3. CFAAs must be soaked in water for an hour and then dried with a towel. In a concrete mixer, these aggregates were mixed with cementitious materials and sand for 30 s. After adding water and superplasticizer, the mixture continued to be stirred for 150 s. The resulting fresh concrete was poured into molds to prepare the cubic specimens (100 mm × 100 mm ×100 mm) and the prismatic specimens (100 mm × 100 mm × 400 mm). In order to prevent the aggregates from floating in the mortar, the rodding method was adopted to compact the concrete mixture. After 24 h, all specimens were demolded and marked with identification codes. The specimens would be moved into a water storage tank within 30 min and cured at ambient temperature until the required age.

### 2.4. *Saline Solution Preparation*

In the western Jilin Province of China, there is a mass of sulfate ions and carbonate ions in the soil. The concentration of these ions ranges from 2.4 wt% to 5.1 wt% [50]. In this experiment, $Na_2SO_4$ solution and $Na_2CO_3$ solution were prepared with distilled water. The molar concentration of the solutions was set to 0.35 mol/L to ensure that the experimental conditions could simulate the actual situation in saline soil areas.

### 2.5. *Test Methods*

The strength tests and microscopic tests of CFAA concrete were conducted to determine the optimum NS dosage. The concrete with the optimum NS dosage was denoted as NOP. Subsequently, the durability in a salt-corrosion environment was studied through dry–wet and freeze–thaw cycle tests. The test procedure of the specimens is plotted in Figure 4.

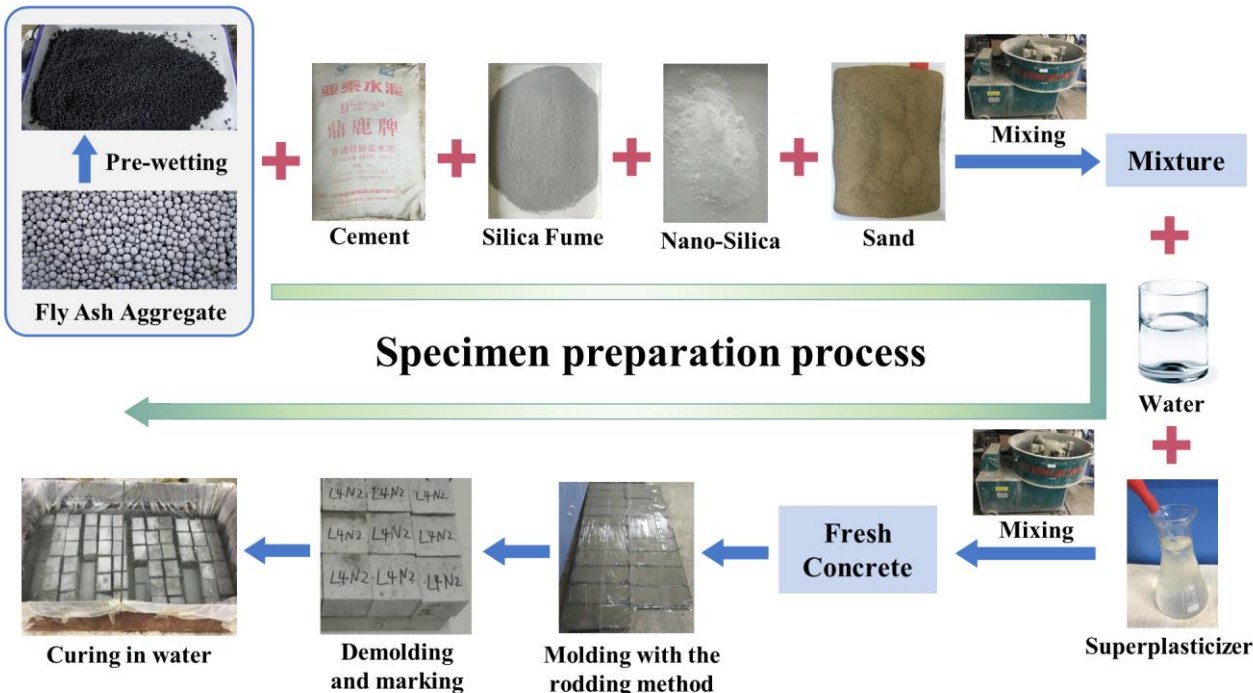

**Figure 3.** Preparation process of CFAA concrete specimens (the non-English characters appearing above the word "Cement" represent the brand of cement).

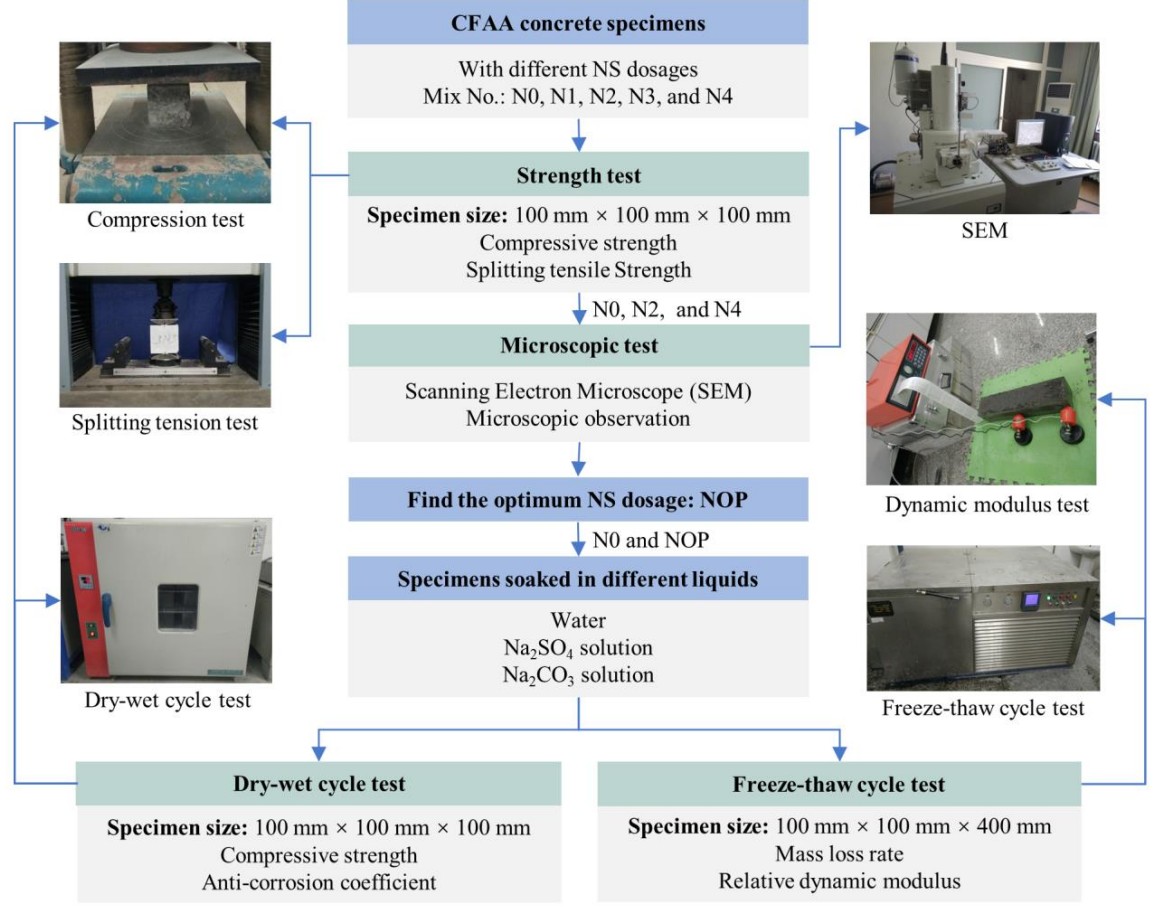

**Figure 4.** Test procedure of CFAA concrete specimens.

### 2.5.1. Compressive Strength Testing

Based on the Chinese standard GB/T 50081-2019 [63], the cubic specimens (100 mm × 100 mm × 100 mm) were utilized to perform the compressive strength tests. The concrete (N0, N1, N2, N3, and N4) contained the specimens at a series of ages (i.e., 3 days, 7 days, 14 days, 28 days, and 56 days). The YAW-2000 compression testing machine (Changchun New Testing Machine Co., Ltd., Changchun, China) was used in all compression tests. The loading rate was set to 0.5 MPa/s.

### 2.5.2. Splitting Tensile Strength Testing

The splitting tensile tests of the cubic specimens (100 mm × 100 mm × 100 mm) were carried on according to the standard GB/T 50081-2019 [63]. After 28 days of curing, the concrete (N0, N1, N2, N3, and N4) was tested using the DNS300 electronic universal testing machine (Changchun Research Institute for Mechanical Science Co., Ltd., Changchun, China) at a loading rate of 0.05 MPa/s.

### 2.5.3. Microscopic Testing

After the strength tests, three specimens, N0, N2, and N4, were made into SEM samples to observe the microstructure of the cementitious matrix.

### 2.5.4. Dry–Wet Cycle Testing

The cubic specimens (100 mm × 100 mm × 100 mm) of the concrete (N0 and NOP) were prepared to conduct dry–wet cycle tests. Based on the Chinese standard GB/T 50082-2009 [64], the process of the tests was the following. The 26-day specimens were taken out from the water storage tanks and wiped dry. The 101-3A electric blast drying oven (Shanghai Yetuo Technology Co., Ltd., Shanghai, China) was utilized to dry these specimens for 48 h at 80 °C (±5°C). After the specimens were cooled to the ambient temperature, the first dry–wet cycle was performed. To begin with, the cubes (N0 and NOP) were fully immersed in the distilled water, the $Na_2SO_4$ solution, and the $Na_2CO_3$ solution, respectively. After a soak of 15 h, the cubes were lifted out of the liquids and air-dried for an hour. Subsequently, the specimens were dried for 6 h at 80 °C (±5°C) using the oven and then cooled off for 2 h to 25–30 °C. In the end, these cubes were soaked in the liquids again to continue with the next dry–wet cycle. The steps mentioned above made up a 24 h dry-wet cycle. There were 60 dry–wet cycles in this experiment. Every 10 cycles, the cubes were removed from the liquids to test the compressive strength.

The anticorrosion coefficient $K_f$ evaluates the salt-corrosion resistance of the specimens and is expressed as follows:

$$K_f = \frac{f_{cns}}{f_{cnw}} \tag{1}$$

where $f_{cns}$ and $f_{cnw}$ denote the compressive strength of the specimens subjected to $n$ dry–wet cycles in saline solution and water, respectively.

### 2.5.5. Freeze–Thaw Cycle Testing

The rapid freezing method was adopted in freeze–thaw cycle tests in accordance with the standard GB/T 50082-2009 [64]. For the concretes N0 and NOP at 28 days, the prismatic specimens (100 mm × 100 mm × 400 mm) were moved into the TDRF-1 concrete rapid freezing and thawing test machine (Hangzhou Civit Instrument Equipment Co., Ltd., Hangzhou, China). Before the freeze–thaw cycles started, these specimens needed to be immersed in water, $Na_2SO_4$ solution, and $Na_2CO_3$ solution. Every liquid contained three N0 specimens and three NOP specimens. This experiment included 150 freeze–thaw cycles in total. Every 10 cycles, all specimens were taken out to acquire their mass (YP3001 electronic scale) and transverse fundamental frequency (DT-20 dynamic modulus tester). After the measurement, these specimens were put back to continue the cycle tests.

The mass loss rate $\Delta W_n$ represents the erosion degree of the specimen surface and is given as follows:

$$\Delta W_n = \frac{W_0 - W_n}{W_0} \times 100\% \tag{2}$$

where $W_0$ and $W_n$ are the initial mass of specimens and the mass after $n$ freeze–thaw cycles, respectively.

The relative dynamic modulus $P$ estimates the internal damage of the specimens and is defined as follows:

$$P = \left(\frac{v_n}{v_0}\right)^2 \times 100\% \tag{3}$$

where $v_0$ and $v_n$ denote the initial transverse fundamental frequency of specimens and the transverse fundamental frequency after $n$ freeze–thaw cycles, respectively.

## 3. Results and Discussion

### 3.1. Mechanical Properties

#### 3.1.1. Compressive Strength

Figure 5 presents the compressive strength and strength growth rate of CFAA concrete containing different NS dosages. The compressive strength of the concrete rises with increasing curing time. For the concrete at the same age, the strength growth rates peak at 2 wt% NS dosage. In the concrete, NS not only reacts with Calcium Hydroxide (CH) to form extra C-S-H gels but also fills nanoscale pores in the cementitious matrix [54]. The compressive strength would increase due to the strengthened matrix when the NS dosage varies from 0 to 2 wt%. Note that replacing cement partially with NS can save cement but lead to the less production of the CH resulting from cement hydration. When the CH is used up, the excess NS scarcely continues to produce C-S-H gels, reducing the total amount of the C-S-H gel. Meanwhile, the excess NS particles might agglomerate together to form weak-zones in the concrete [65]. Therefore, the compressive strength would decrease when the NS dosage exceeds 2 wt%. This result shows the optimum NS dosage of the concrete is 2 wt%, i.e., N2 is the NOP in Section 2.5.

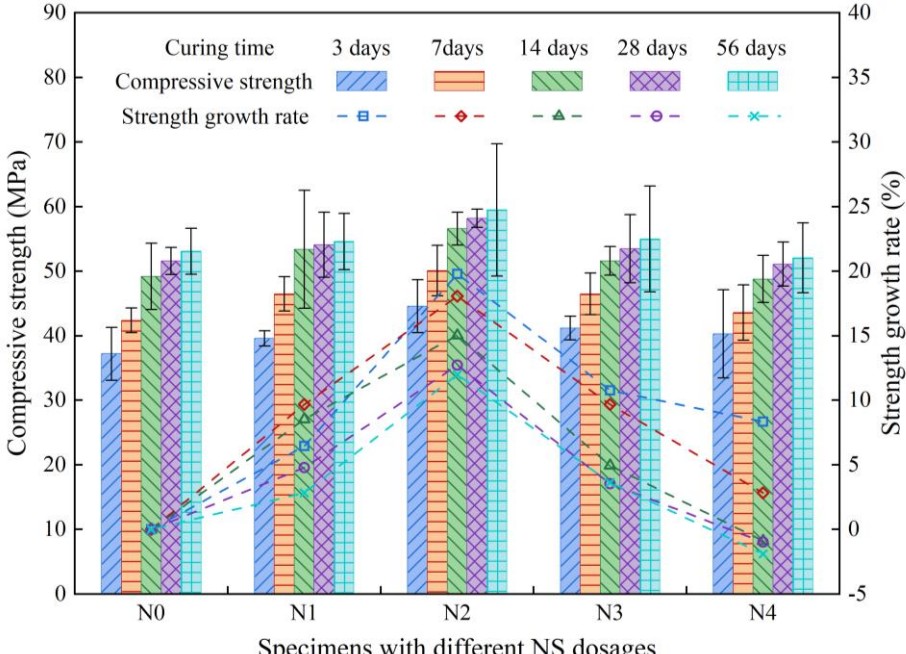

**Figure 5.** Compressive strength and strength growth rate of CFAA concrete containing different NS dosages.

As seen from Figure 5, the compressive strength of the 3-day concrete (N1, N2, N3, and N4) increased by 6.5%, 19.8%, 10.8%, and 8.3%, respectively, compared to N0. At 7 days, the strength growth rates were 9.7%, 18.1%, 9.7%, and 2.8%, respectively. After 14 days of curing, the growth of the strength started to slow down significantly, and the corresponding growth rates were 8.5%, 15.0%, 5.0%, and −0.9%, respectively. At 28 days, the strength growth rates of N1, N2, N3, and N4 further dropped to 4.8%, 12.7%, 3.6%, and −1.0%, respectively. The 56-day strength growth rates were 2.8%, 12.0%, 3.6%, and −1.9% for N1, N2, N3, and N4, respectively, and approximated to the 28-day strength growth rates. It can be found that the early-age strength growth rate was higher than the late-age strength growth rate for NS-modified CFAA concrete. The 3-day strength growth rate is the highest and up to 19.8% at 2 wt% NS dosage. This is because the NS particle act as a crystal nucleus of C-S-H gels in the early hydration stage [54]. At the same time, the pozzolanic reaction of NS consumes the CH and thus promotes cement hydration. These two reasons would accelerate the production of the C-S-H gels in the early stage and improve significantly the early compressive strength of the concrete.

Moreover, the strength growth rates of the concrete at 14 days, 28 days, and 56 days are less than zero when the NS dosage is 4 wt%. Under this condition, the concrete without NS had higher compressive strength, indicating that the excess NS is harmful to the concrete properties. It is recommended that the NS dosage should be determined by experiment when NS is used as a mineral admixture.

### 3.1.2. Splitting Tensile Strength

Figure 6 illustrates the splitting tensile strength and strength growth rate of CFAA concrete containing different NS dosages. Compared with N0 concrete, the splitting tensile strength of N1, N2, N3, and N4 increases by 7.8%, 11.2%, 4.1%, and −0.8%, respectively. This result shows that the splitting tensile strength is the highest at 2 wt% NS dosage and decreases when the NS dosage varies from 2 wt% to 4 wt%. This is because the NS with a moderate dosage produces more C-S-H gels through the pozzolanic reaction to fill pores in the cementitious matrix. The splitting tensile strength of the concrete improves as the matrix porosity decreases. On the contrary, the excess NS increases the number of pores and weak-zones in the matrix, reducing the splitting tensile strength.

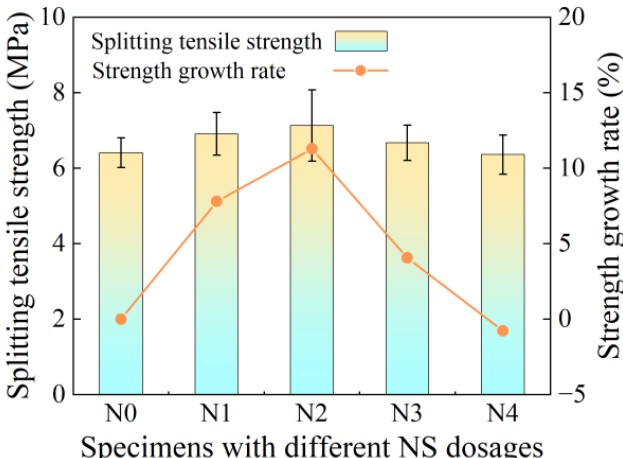

**Figure 6.** Splitting tensile strength and strength growth rate of CFAA concrete containing different NS dosages.

Based on the results above and the investigations [5,6], the relationship between splitting tensile strength and compressive strength for 28-day CFAA concrete can be obtained, as shown in Figure 7. Considering that the experiments had different specimen sizes, all data had been converted to the strength values of 100 mm cubes according to the standard

GB/T 50081-2019 [63]. Through regression analysis, the equation of the fitting line can be expressed as follows:

$$y = -0.809 + 0.14x \tag{4}$$

where $x$ and $y$ represent the compressive strength and the splitting tensile strength, respectively. The coefficient of determination, $R^2$, is 0.976, indicating that the equation has high goodness of fit. It can be found that the splitting tensile strength increases with the growth of the compressive strength, and their relationship is approximately linear. Note that the relationship is applicable to the cubic specimens (100 mm × 100 mm × 100 mm) with the nominal aggregate size grade of 5–10 mm. When the specimen size and aggregate gradation change, this relationship must be investigated further.

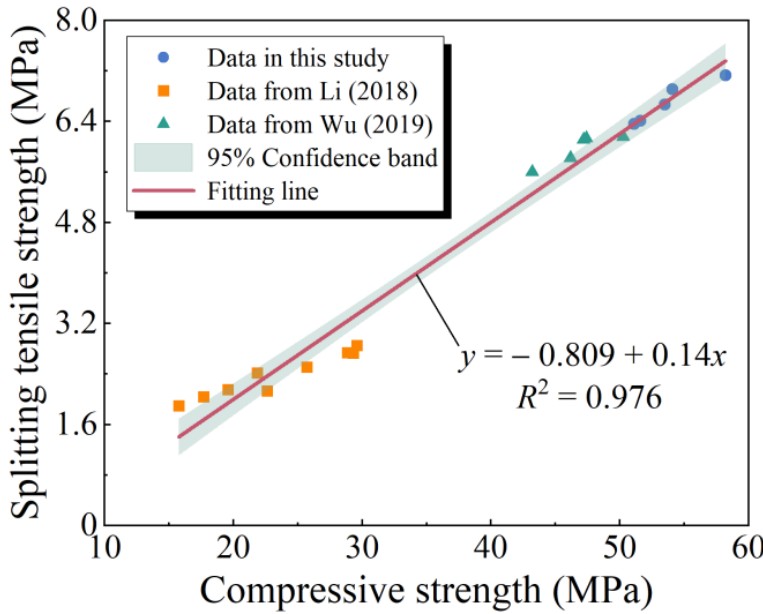

**Figure 7.** Relationship between splitting tensile strength and compressive strength for 28-day CFAA concrete (some data were sourced from Li [5] and Wu [6]).

### 3.1.3. Failure Morphology and SEM Analysis

The failure morphologies of CFAA concrete specimens in strength tests are shown in Figure 8. The CFAA concrete made with or without NS had similar failure morphology. During the compression tests, the top and bottom surfaces of the cubes are confined by friction. The initial cracks would occur at every corner of the cubes due to stress concentration. These cracks develop diagonally and converge in the middle of the cubes, resulting in severe concrete spallation from the lateral sides. The shape of the remaining concrete is close to the combination of two cones with opposing cone tops, as presented in Figure 8a. In the splitting tensile tests, the cubes are directly divided into two parts, and the fracture surface is approximately parallel to the surface of the cubes, as illustrated in Figure 8b.

Figure 8c depicts the details of the failed specimen after the compression testing. The result shows that the CFAAs are uniformly distributed on the fracture surface. It can also be observed that some CFAAs remain intact, and the interfacial debonding occurs between the aggregates and the matrix. The reason is that the aggregates have a spherical appearance and a smooth surface, leading to low interfacial strength. However, other CFAAs are crushed, and the damaged aggregates mainly concentrate in moist areas, as shown in Figure 8c. This may be because the moist areas resulting from the water penetration contain adequate water to ensure full cement hydration. The bonding between the aggregates and the matrix would be enhanced. Furthermore, the aggregates consist of C-S-H gels and have a similar strength to the matrix. Consequently, the fracture surface might penetrate

the aggregates when the interfacial strength is enhanced. In conclusion, the compressive strength of the concrete is related to the aggregate strength, the matrix strength, and the bonding strength between the aggregates and the matrix. These factors also influence the splitting tensile strength of lightweight concrete [18,66,67]. This can explain why the strong correlation exists between the splitting tensile strength and the compressive strength of CFAA concrete in Section 3.1.2.

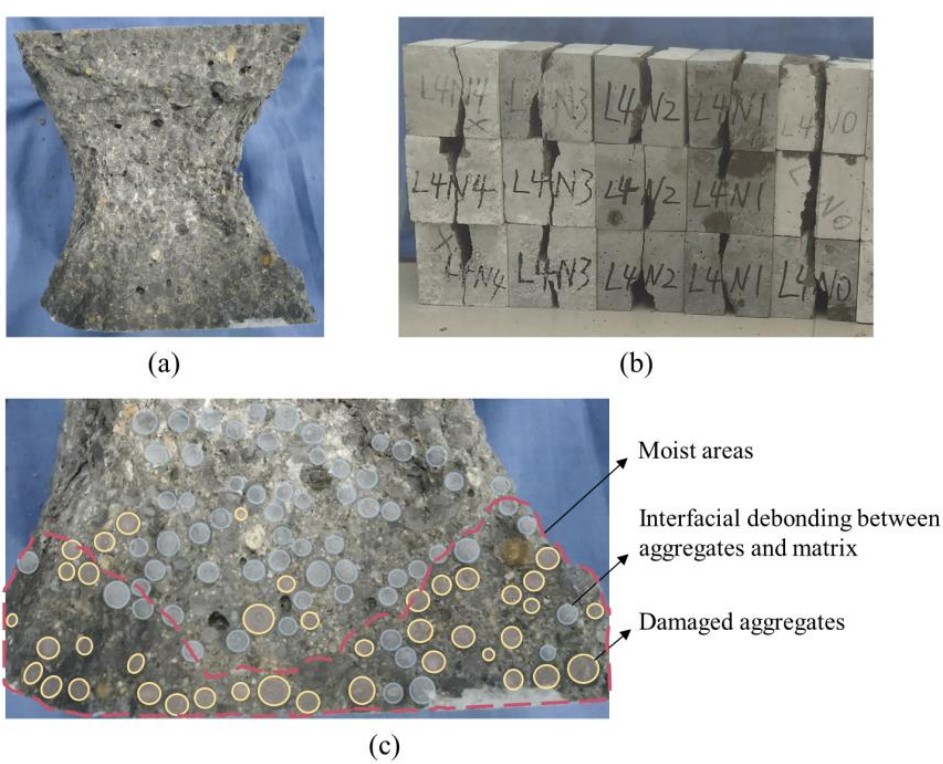

**Figure 8.** Failure morphologies of CFAA concrete specimens in strength tests: (**a**) after compression testing; (**b**) after splitting tensile testing; (**c**) details of the failed specimen after the compression testing.

After 28 days of curing, the three most representative cubes, N0, N2, and N4, were made into SEM samples to observe their microstructure. The SEM images of the CFAA concrete (N0, N2, and N4) are presented in Figure 9. There are many pores among the C-S-H gels in the concrete without NS, as illustrated in Figure 9a. With the addition of 2 wt% NS, the number and size of pores decrease greatly, and the C-S-H gels become more homogeneous, as shown in Figure 9b. The reason is that NS reacting with CH produces the additional C-S-H gels to fill the big holes so that the matrix is enhanced. At the same time, NS can fill the nanoscale pores in the matrix to make the microstructure much denser. However, when the NS dosage increases to 4 wt%, pores and microcracks would arise in the matrix so that the microstructure gets loose, as depicted in Figure 9c. Compared with N0, larger pores are observed in N4. Furthermore, the initiation of microcracks might relate to the agglomeration of NS particles because the drying shrinkage of concrete can contribute to forming the microcracks around the agglomerated NS particles [37]. Overall, the moderate NS significantly optimizes the microstructure of the matrix, whereas the excess NS causes more microscale defects in the matrix.

### 3.2. Durability Properties

The strength tests in Section 3.1 show that the optimum NS dosage is N2. During the durability tests, the specimens, N0 and N2, were soaked in different liquids (water, $Na_2SO_4$ solution, and $Na_2CO_3$ solution). The dry–wet and freeze–thaw cycle tests were conducted to explore the durability of CFAA concrete.

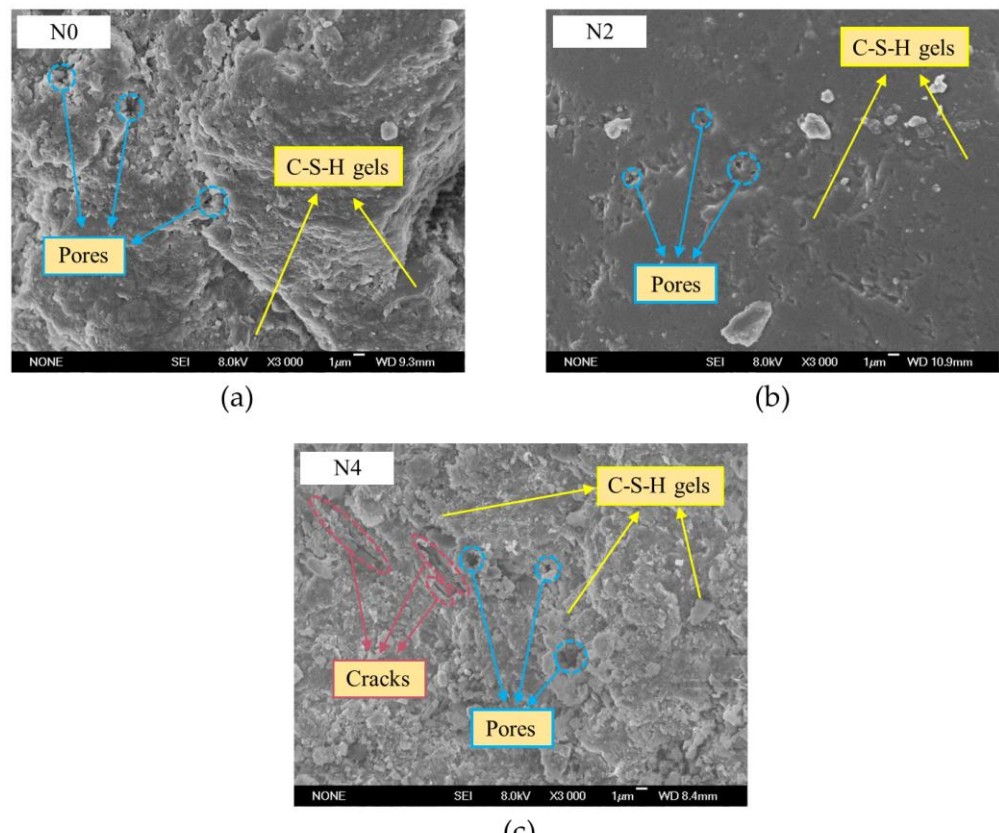

**Figure 9.** SEM images of 28-day CFAA concrete containing different NS dosages: (**a**) N0; (**b**) N2; (**c**) N4.

### 3.2.1. Results of the Dry–Wet Cycle Testing

The compressive strength of N0 and N2 subjected to the dry–wet cycles in water are listed in Table 4. The compressive strength slightly rises as the number of dry–wet cycles increases. This is because, in the wet period of each cycle, the cement continues to react with water so that the concrete is enhanced. To eliminate the influence of cement hydration on the strength, the results in Table 4 were used to calculate the anticorrosion coefficient through Equation (1).

**Table 4.** Compressive strength of CFAA concrete subjected to dry–wet cycles in water.

| Dry–Wet Cycles | | 0 | 10 | 20 | 30 | 40 | 50 | 60 |
|---|---|---|---|---|---|---|---|---|
| N0 | Mean (MPa) | 51.93 | 52.40 | 52.40 | 52.50 | 52.97 | 53.37 | 53.57 |
| | COV [1] (%) | 5.90 | 4.58 | 8.51 | 9.86 | 5.32 | 7.16 | 8.60 |
| N2 | Mean (MPa) | 56.03 | 56.57 | 57.03 | 57.07 | 57.20 | 57.80 | 57.83 |
| | COV [1] (%) | 6.52 | 2.37 | 5.10 | 4.44 | 4.85 | 5.50 | 1.44 |

[1] COV denotes the coefficient of variation.

Figure 10 shows the compressive strength and the anticorrosion coefficient of the CFAA concrete subjected to the dry–wet cycles in the $Na_2SO_4$ and $Na_2CO_3$ solutions. It can be found that the compressive strength of N2 is always higher than that of N0 due to the pozzolanic reaction and filler effects of NS. Moreover, the standard deviation of these strength values varied from 0.21 to 4.35, and the maximum value was about 20 times the minimum value. The main reason for such a big difference might be the use of the rodding method during the casting of concrete. The rodding method is manually operated and has difficulty in ensuring that the concrete specimens have a similar degree of compaction. The standard deviation of the compressive strength values is generally between 1 and 5 in

this experiment. In rare cases, the degree of concrete compaction in the specimens might be close to each other. This would cause the possibility that the standard deviation could decrease and be significantly less than 1.

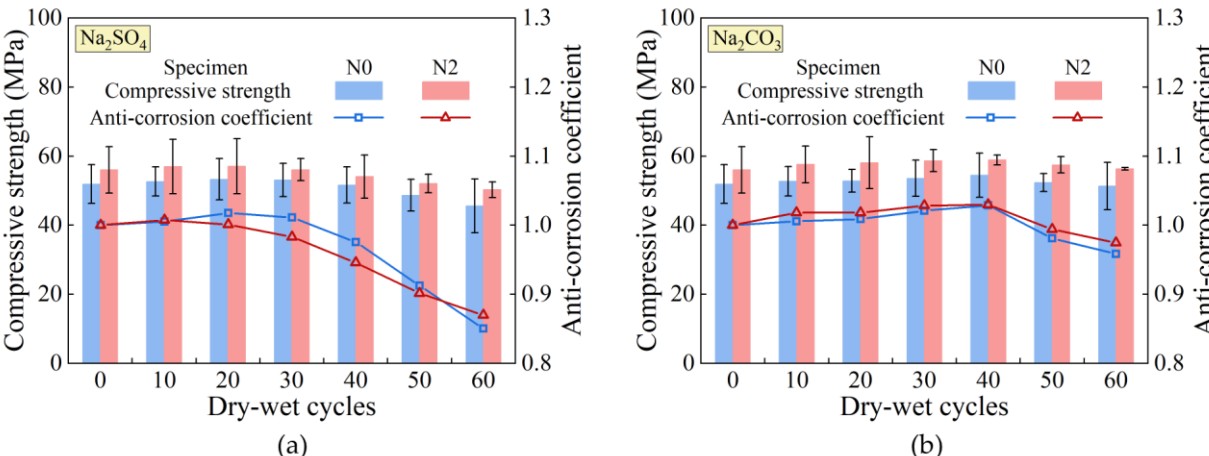

**Figure 10.** Compressive strength and anticorrosion coefficient of CFAA concrete subjected to dry–wet cycles in different saline solutions: (**a**) $Na_2SO_4$ solution; (**b**) $Na_2CO_3$ solution.

As observed from Figure 10, the anticorrosion coefficient slightly rises in the early stage of dry–wet cycles and then significantly decreases. The increase in the anticorrosion coefficient is attributed to the filler effects of the reaction products and the salt crystals. However, when the pore size hinders the growth of the crystals, crystallization pressure arises and causes the formation of microcracks around the pores. Meanwhile, sulfates or carbonates consume the CH and thus destroy the alkaline environment inside the concrete, decreasing the matrix strength. Therefore, crystallization pressure and matrix degradation lead to a reduction in the anticorrosion coefficient. In the $Na_2SO_4$ solution, the anticorrosion coefficient of N0 and N2 reaches the peak at cycles 10 and 20, respectively. In the $Na_2CO_3$ solution, the highest anticorrosion coefficient in N0 and N2 occurs at cycle 40. This indicates that CFAA concrete is more susceptible to erosion in the $Na_2SO_4$ solution than in the $Na_2CO_3$ solution. This is because sulfate ions can react with CH and Calcium-Aluminate-Hydrate (C-A-H) gels to form Ettringite (Et). The volume of the Et is more than 2.5 times that of the original C-A-H gels, resulting in the generation of expansive stress within the concrete. Hence, in the $Na_2SO_4$ solution, the concrete is simultaneously subjected to the crystallization pressure and the internal expansive stress, accelerating the performance degradation.

In Figure 10a, for the anticorrosion coefficient, the peak of N0 occurs later than that of N2 in the $Na_2SO_4$ solution. The reason may be that N0 has more voids than N2. The voids in N0 would take more dry–wet cycles to be filled with the reaction products and crystals. After 40 cycles, the anticorrosion coefficient of N2 falls slower than that of N0 in the $Na_2SO_4$ solution. This is because N2 has a higher matrix strength than N0, contributing to resisting the internal expansive stress and the crystallization pressure. Thus, NS can improve the corrosion resistance of the concrete to sulfates in the late stage of the dry–wet cycle. In Figure 10b, the anticorrosion coefficient of N2 is higher than that of N0 in the $Na_2CO_3$ solution. During cycles 40 to 60, the anticorrosion coefficient decreases by 6.86% and 5.35% in N0 and N2, respectively. This shows that N2 has a stronger corrosion resistance than N0 in the $Na_2CO_3$ solution, especially in the late dry–wet cycles. Consequently, NS can enhance the corrosion resistance of CFAA concrete to carbonates.

### 3.2.2. Results of the Freeze–Thaw Cycle Testing

Figure 11 presents the mass loss rate of the specimens (N0 and N2) under freeze–thaw cycles. During the freeze–thaw cycles, the pore solutions are converted into solids due to

low temperature, causing the frost-heaving force around the pores. When the frost-heaving force exceeds the tensile strength of the matrix, the concrete spallation would arise and gradually get severe. Hence, the mass loss rate rises with the increasing freeze–thaw cycles. In Figure 11, the mass loss rate in saline solutions exceeds that in water after 20 cycles, because in the thaw period of the cycles, some reaction products and salts are taken out of the concrete as the pore water moves, increasing the porosity of surfaces. The concrete in saline solutions would have much severer surface spallation than in water during the freeze–thaw cycles. Meanwhile, the mass loss rate in the $Na_2SO_4$ solution is higher than that in the $Na_2CO_3$ in the late freeze–thaw cycles. Under sulfate attacks, C-A-H gels are consumed to generate the expansive Et, dramatically reducing the matrix strength near surfaces. For this reason, concrete spallation would be severer in the $Na_2SO_4$ solution than in the $Na_2CO_3$ solution with the growing freeze–thaw cycles.

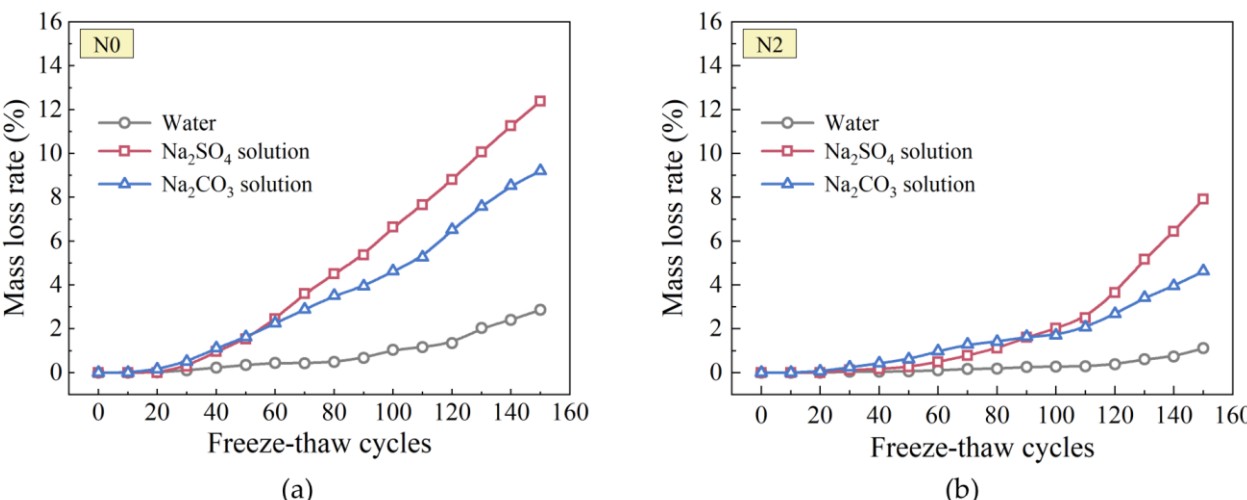

**Figure 11.** Mass loss rate of the specimens under freeze–thaw cycles: (**a**) N0; (**b**) N2.

In Figure 11a,b, at cycle 150, the mass loss rate of N2 in water, $Na_2SO_4$ solution, and $Na_2CO_3$ solution, compared with N0, decreases by 61.2%, 36.1%, and 49.7%, respectively. This result indicates that the CFAA concrete has stronger surface spallation resistance after incorporating NS. The reason is that the pozzolanic effect of NS enhances the matrix strength, contributing to resisting the frost-heaving force. Therefore, NS can improve the surface spallation resistance of CFAA concrete during freeze–thaw cycles.

Figure 12 illustrates the relative dynamic modulus of the specimens (N0 and N2) under freeze–thaw cycles. In addition to the frost-heaving force, the osmotic pressure resulting from the concentration difference of pore solutions would arise in the freezing stage of the cycles. The osmotic pressure and the frost-heaving force cause the continued growth of microcracks around the pores, increasing the internal damage of the concrete. For this reason, the relative dynamic modulus decreases as the number of freeze–thaw cycles grows. In Figure 12, the concrete in saline solutions has a significantly higher relative dynamic modulus than that in water in the late stage of the freeze–thaw cycles. This is because the saline solution has a lower freezing point than water. At the same temperature, the frost-heaving force in saline solution is weaker compared with water, resulting in less internal damage. Moreover, the relative dynamic modulus in the $Na_2SO_4$ solution is gradually lower than that in the $Na_2CO_3$ solution after 30 cycles. The reason is that sulfate attacks can destroy the C-A-H gels and generate the expansive Et. In consequence, the decreased matrix strength and the increased internal stress accelerate the internal damage of the concrete.

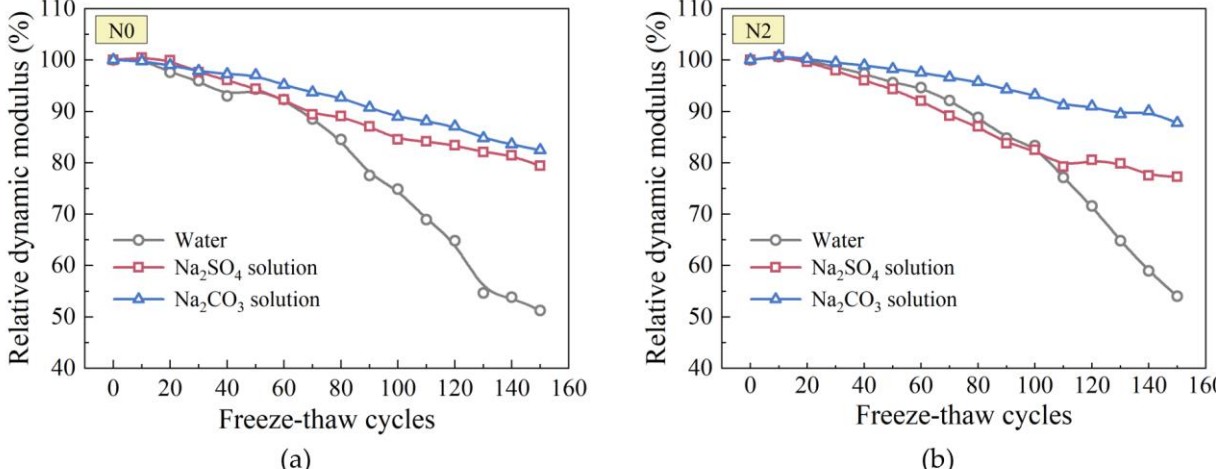

**Figure 12.** Relative dynamic modulus of the specimens under freeze–thaw cycles: (**a**) N0; (**b**) N2.

In Figure 12a,b, at cycle 150, compared with N0, the relative dynamic modulus of N2 in water, $Na_2SO_4$ solution, and $Na_2CO_3$ solution increases by 5.4%, −2.7%, and 6.5%, respectively. The matrix has fewer pores and higher strength with the addition of NS, contributing to resisting the osmotic pressure and the frost-heaving force. Thus, NS can reduce the internal damage of CFAA concrete during freeze–thaw cycles in the water and the $Na_2CO_3$ solution. However, N2 has a lower relative dynamic modulus than N0 in the $Na_2SO_4$ solution. This exhibits the fact that NS can hardly improve the internal frost resistance of the concrete subjected to sulfate attacks.

## 4. Conclusions

To obtain satisfactory long-term performance, Nanosilica (NS)-modified Cold-bonded Fly Ash Aggregate (CFAA) concrete was prepared to conduct strength tests and microstructure analysis. The durability was evaluated by dry–wet and freeze–thaw cycle tests in different saline solutions. The main conclusions are drawn as follows:

1.  After incorporating NS, the compressive and splitting tensile strength of CFAA concrete is enhanced, and the optimum NS dosage is 2 wt%. NS also plays a crucial role in improving the early-age strength of the concrete.
2.  The strength of CFAA concrete is related to the aggregate strength, the matrix strength, and the bonding strength between the aggregate and the matrix. NS can reduce the porosity and improve the properties of the cementitious matrix, whereas excess NS would cause more microscale deficiencies in the matrix.
3.  During dry–wet cycles, CFAA concrete has stronger corrosion resistance to carbonates than to sulfates. The addition of NS effectively alleviates the concrete deterioration in the late dry–wet cycles.
4.  Under freeze–thaw conditions, NS-modified CFAA concrete exhibits desirable spallation resistance and internal frost resistance. However, NS could barely diminish the internal damage due to sulfate attacks.

This paper focuses on utilizing NS to enhance the strength and durability of CFAA concrete. The research results facilitate alleviating the concrete deterioration under salt attacks, offering a reference for the potential application of the concrete in a harsh environment. Further studies would benefit from performing the quantitative analysis of the existing data to establish damage models for CFAA concrete. It would also be valuable to investigate the long-term performance of the concrete subjected to the coupling action of sulfates and carbonates. Several factors could be considered in experiments, such as the NS dosage, the salt concentration, and the ratio of sulfates to carbonates.

**Author Contributions:** Conceptualization, Q.W.; investigation: J.W.; methodology, J.W.; supervision, Q.W.; visualization: X.P.; writing—original draft preparation, X.P.; writing—review and editing, X.P. All authors have read and agreed to the published version of the manuscript.

**Funding:** This research received no external funding.

**Institutional Review Board Statement:** Not applicable.

**Informed Consent Statement:** Not applicable.

**Data Availability Statement:** The authors confirm that the data supporting the findings of this study are available within the article.

**Conflicts of Interest:** The authors declare no conflict of interest.

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
