# Peer review of "Effect of Nanosilica on the Strength and Durability of Cold-Bonded Fly Ash Aggregate Concrete"

_sustainability, doi:10.3390/su152115413_

Round 1

Reviewer 1 Report

Comments and Suggestions for Authors

Dear Authors, can you please enrich your introduction chapter with more detailed information and references? Can you please explain why in Fig.10 you have such big difference in stdev values among samples. Your conclusion part must be detailed explained on the outcome of your work and further recommendations.

Comments on the Quality of English Language

Ok

Reviewer 2 Report

Comments and Suggestions for Authors

The article on “Effect of nano-silica on the strength and durability of cold-bonded fly ash aggregate concrete” is well written. However, the following comments/suggestions have to be addressed to improve the manuscript.

1.      More literature should be included in the introduction chapter to substantiate the novelty and need of the study.

2.      Research significance or highlights of the study should be provided in bulletin points at the end of introduction chapter

3.      It is strongly recommended to provide subheadings of each raw materials under section 2.1 for better understanding.

4.      The particle gradation of CFAA should be essentially provided.

5.      The size of CFAA used is stated as 3 to 5 mm pertains to fine aggregate size ranges. How can it be considered as coarse aggregate?

6.      The percentage improvement in the mechanical strength properties with the age and dosage of NS should be elaborated further in the discussions.

1.      Recommendation from the output of the study should be provided at the end of conclusion

Reviewer 3 Report

Comments and Suggestions for Authors

The authors used the experimental-based method to evaluate the effect of Nano-Silica on the durability of cold-bonded fly ash aggregate concrete in a salt corrosion environment. The methodology was described comprehensively and the results are meaningful and valuable. However, there are some problems existing in this paper that the authors must pay attention to deal with.

1.       The background and significance of this study should be highlighted in the abstract.

2.       In the Introduction, please add the background knowledge of cold-bonded fly ash aggregate concrete.

3.       Some expressions in the manuscript are not clear. An English native speaker is suggested to proofread it again carefully.

4.       Please provide the particle size sieving curve of CFAA to check if it meets the requirements for coarse aggregate.

5.       What are the criteria for selecting the NS dosages?

6.       Please add the mix proportions of N4 CFAA concrete.

7.       Some mistakes can be found as well. Please carefully check through the manuscript. e.g., in Fig. 4, the mass loss rate.

8.       There are too few fitting data points in Fig. 7.

Comments on the Quality of English Language

Some expressions in the manuscript are not clear. An English native speaker is suggested to proofread it again carefully.

Round 2

Reviewer 2 Report

Comments and Suggestions for Authors

The comments/suggestions are well addressed